# Dose–Response Effects of Glutathione Supplement in Parenteral Nutrition on Pulmonary Oxidative Stress and Alveolarization in Newborn Guinea Pig

**DOI:** 10.3390/antiox11101956

**Published:** 2022-09-30

**Authors:** Jean-Claude Lavoie, Ibrahim Mohamed, Vitor Teixeira

**Affiliations:** 1Research Center of the CHU Sainte-Justine, Department of Nutrition, Université de Montréal, Montréal, QC H3T 1C5, Canada; 2Research Center of the CHU Sainte-Justine, Department of Paediatrics, Université de Montréal, Montréal, QC H3T 1C5, Canada

**Keywords:** bronchopulmonary dysplasia, chronic lung disease, glutathione, glutathione supplementation, oxidative stress, parenteral nutrition, peroxide, premature newborn, pulmonary alveolarization, redox potential

## Abstract

In premature infants, glutathione deficiency impairs the capacity to detoxify the peroxides resulting from O_2_ metabolism and those contaminating the parenteral nutrition (PN) leading to increased oxidative stress, which is a major contributor to bronchopulmonary dysplasia (BPD) development. In animals, the supplementation of PN with glutathione prevented the induction of pulmonary oxidative stress and hypoalveolarization (characteristic of BPD). Hypothesis: the dose of glutathione that corrects the plasma glutathione deficiency is sufficient to prevent oxidative stress and preserve pulmonary integrity. Three-day-old guinea pigs received a PN, supplemented or not with GSSG (up to 1300 µg/kg/d), the stable form of glutathione in PN. Animals with no handling other than being orally fed constituted the control group. After 4 days, lungs were removed to determine the GSH, GSSG, redox potential and the alveolarization index. Total plasma glutathione was quantified. The effective dose to improve pulmonary GSH and prevent the loss of alveoli was 330 µg/kg/d. A 750 µg/kg/d dose corrected the low-plasma glutathione, high-pulmonary GSSG and oxidized redox potential. Therefore, the results suggest that, in a clinical setting, the dose that improves low-plasma glutathione could be effective in preventing BPD development.

## 1. Introduction

Peroxides contaminating parenteral nutrition [1,2,3,4,5] are suspected to be a major trigger for the development of bronchopulmonary dysplasia (BPD) in extreme premature newborns [5,6,7]. This chronic lung disease affects 30–60% of premature infants born before 30 weeks of gestation [8]. Newborn guinea pigs infused with parenteral nutrition (PN) or a solution containing peroxides at a concentration measured in PN exhibited an oxidized redox potential of glutathione [9,10,11], exaggerated apoptosis [3,9] and hypoalveolarization [3,9,10,11], the main characteristics of BPD. Increased urinary peroxides in the first week of life of preterm infants were associated with BPD or death [5]. The association between oxidized redox potential and BPD is also reported [6,12].

An oxidized redox potential results from an imbalance between the two forms of glutathione: GSH (reduced) and GSSG (oxidized). This could occur during the detoxification of high levels of peroxides by the action of glutathione peroxidases (GPx). During the process, GSH is transformed into GSSG. In the presence of high levels of peroxides, the rate of GSSG production could be greater than the capacity of glutathione reductase (GR) to recycle GSSG into GSH. It is, therefore, important to detoxify the infused peroxides upon their arrival in cells, thereby avoiding their accumulation [5], and a subsequent rise in GSSG levels, which is associated with oxidized redox potential. In premature infants, the GPx activity measured in the blood seems normal (similar to those observed in adults [5,13]), while the total glutathione in the plasma is approximately 25% that of normal levels [5]. Thus, insufficient GSH levels are suspected in the lungs of premature infants, limiting the ability of GPx to detoxify peroxides. In animals, the addition of GSSG as a pro-cysteine in PN allowed for plasma glutathione to be increased and prevented oxidation of the pulmonary redox potential, exaggerated apoptosis and loss of alveoli [10]. Based on these data, a Phase 1 clinical study is considered. However, the preclinical study protocol [10] included only one dose of glutathione. The purpose of the present study was to generate data from a preclinical dose–response study to qualify the effective dose to correct plasma glutathione deficiency, prevent pulmonary oxidative stress (characterized by the redox potential of glutathione) and preserve pulmonary integrity.

The infused peroxides must reach a toxic level before an increase is observed in oxidative stress, an exaggerated apoptosis and a loss of alveoli [10]. Indeed, the photo-protection of PN, which halves the peroxide level [14], prevented these injuries in the animals [10]. Additionally, Mohamed’s study [5] on the link between BPD and urinary ascorbylperoxide concentration reported that premature infants who were unable to detoxify peroxides developed BPD. This study shows that the level of peroxide increases over time, reaching toxic levels. Therefore, the hypothesis of the present study was that the infusion of GSSG works to reduce peroxides in the lungs to a non-toxic threshold. This dose of GSSG would be lower than that necessary to eliminate all the peroxides, and thus protect the plasma pool of glutathione. Our results confirm this hypothesis. In our animal model, a dose of 330 µg/kg/d was effective in protecting the lung structure, while a dose of 750 µg/kg/d was required to normalize plasma glutathione levels. Therefore, in a clinical setting, the dose that corrects the plasma deficiency in glutathione is expected to be effective in preventing the development of BPD.

## 2. Materials and Methods

### 2.1. Experimental Design

The newborn guinea pig has been used for several years as an animal model to study oxidative stress induced by parenteral nutrition. The modification of the redox potential of glutathione allows for a biochemical quantification of the oxidative stress, whereas the pulmonary alveolarization index is a biological manifestation of this oxidative stress. This is, therefore, a relevant model for testing the hypothesis of the efficacy of glutathione supplementation of PN on the reduction in pulmonary oxidative stress induced by PN.

At three days of age, male Hartley Guinea Pigs from Charles River Laboratories (Saint-Constant, QC, Canada) were received by the animal facility of CHU Sainte-Justine. On the day of their arrival, 48 animals were anaesthetized with ketamine/xylazine to insert and fix a catheter (SAI Infusion technologies, Village of Lake Villa, IL, USA) into their right jugular vein, as previously reported [3,9,10,11]. Parenteral nutrition was infused through this catheter for 4 days. At the age of 7 days, all animals were sacrificed for lung and blood sampling, as previously described [11]. The right lung was isolated after ligation of the main bronchus, which was aliquoted and prepared for the determination of glutathione. The left lung was filled with 10% (v,v) paraformaldehyde in PBS [11], isolated and stored in same solution. The histological preparations (hematoxylin-eosin) were carried out by the pathology services of the hospital.

Animals were exclusively on parenteral nutrition (PN): 17.4 g/kg/d dextrose, 4 g/kg/d amino acids (Primene, Baxter, Toronto, ON, Canada); 3.2 g/kg/d lipid emulsion (Intralipid 20%, Pharmacia Upjohn, Baie d’Urfé, QC, Canada); 2 mL/kg/d multivitamin preparation (multi-12 pediatric, Sandoz Canada, Boucherville, QC, Canada) + 1 U/mL heparin and electrolytes. The PN solution, continuously infused at a mean of 190 mL/kg/d, was changed daily.

The disulphide form of glutathione (GSSG) was first added to the parenteral solution bag. GSSG was used rather than GSH due to its improved stability in PN [15] and similar affinity for γ-glutamyl transferase [16]. A recent article [17] demonstrated that the loss of GSSG in PN is caused by the presence of cysteine in PN. The reaction leads to the generation of a mixed disulfide cysteine–glutathione. This molecule is recognized as pro-cystine in vivo, which can be used to increase the synthesis of glutathione in the tissues. γ-glutamyl transferase is involved in the first step of the hydrolysis of plasma glutathione in its constituent amino acids. These amino acids are released into the cells for the de novo synthesis of glutathione. This is how the addition of glutathione to PN improves the lungs’ glutathione level [10]. The final concentration of GSSG in PN varied up to 12 µM (6 groups). The concentrations tested were from a dilution in the PN of a 1 mM stock solution (courtesy of Sandoz Canada). The doses are indicated in µg GSSG/kg/d. In order to limit the impact of batch-to-batch variation of guinea pigs, which are outbreeding animals, during experiment, dose selection was randomized for each group of 4 animals. The doses reported on the abscissa are presented as mean ± S.E.M. according to the exact volume of PN infused into each animal.

Ten other animals served as a reference (control) group. These animals underwent no further manipulation other than regular nutrition for 4 days (until 7 days of age) with standard guinea pig food (Teklad Global High Fiber Guinea Pig Diet, Envigo, Madison, WI, USA).

The protocol was approved by the CHU Sainte-Justine Institutional Committee for Good Practice with Animals in Research, with the principles of the Canadian Council on Animal Care in science.

### 2.2. Determinations

The alveolarization index [3,9,10,11] is defined as the number of intercepts between a calibrated line (1 mm) and histological structures of lung (200X); the higher the index, the higher the number of alveoli. The reported index for each animal was the average value of four different areas of one square millimeter.

Both forms of glutathione (GSH and GSSG) were isolated by capillary electrophoresis, detected at 200 nm and quantified by standard curves of GSH and GSSG, as previously reported [9,10,11]. Concentrations of GSH and GSSG (assuming lung density = 1 g/mL) were used to calculate the redox potential according to the Nernst equation. The redox potential was expressed in millivolts (mV), whereas the pulmonary GSH and GSSG were reported in nmol/mg protein; protein levels were quantified by the Bradford method using bovine serum albumin as standard curve.

GSH and GSSG concentrations in the plasma are too low to be measured by our electrophoresis technique. Therefore, the method described by Griffith OW [18] was used to measure total glutathione (GSH + GSSG) and was reported as the GSH equivalent.

The activities of glutathione peroxidase (GPx) and glutathione reductase (GR) at Vmax were determined as previously published [19]. The assays are based on the enzymatic use of NADPH, monitored at the wavelength of 340 nm. The activities were expressed as nmol of NADPH used/minute/mg protein.

### 2.3. Statistical Analysis

Data are presented as mean ± S.E.M. Groups were orthogonally compared by ANOVA. Homoscedasticity was validated by the Bartlett’s Chi^2^ test. Natural logarithm transformation was used to meet homoscedasticity in comparisons of GSH and GSSG levels in lungs. For each analyzed parameter, the objective was to use orthogonal comparisons, making it possible to highlight the GSSG dose at the origin of a change in the initial values observed in animals receiving PN without GSSG. This dose was defined as the efficient dose. After finding an effective dose using preliminary analysis, this dose was compared to the higher doses including the control (a plateau was suspected), and all doses below the effective dose were compared to the highest doses, including the effective dose. This comparison made it possible to confirm whether the effective dose had reached a plateau. Thus, for Figure 1 (plasma glutathione), the comparisons were {[(0 vs. 95) vs. 240] vs. 330 µg/kg/d} vs. {(C vs. 1300) vs. 750µg/kg/d}. For Figure 2 (Redox), the comparisons were {[(0 vs. 95) vs. 240] vs. 330 µg/kg/d} vs. {(c vs. 1300) vs. 750µg/kg/d}. For Figure 3 (GSH in lungs), the comparisons were {[(0 vs. 95) vs. 240 µg/kg/d]} vs. {[(C vs. 1300) vs. 750] vs. 330 µg/kg/d}}. For Figure 4 (GSSG in lungs), the comparisons were {[(0 vs. 95) vs. 240] vs. 330 µg/kg/d} vs. {[(C vs. 1300) vs. 750 µg/kg/d]}. Figure 5 was not the object of any statistical comparison. For Figure 6 (Alveolarization index), the comparisons were {[(0 vs. 95) vs. 240 µg/kg/d]} vs. {[(C vs. 1330) vs. 750] vs. 330 µg/kg/d}. No multiple comparisons were performed. The significance was set at *p* < 0.05.

## 3. Results

The body weights (110 ± 2 g) did no differ between groups at baseline. At the end of experimentations, body weights did not differ between groups on PN (105 ± 2 g; F_(1,49)_ < 1.6), but were lower than control group for same age (119 ± 6 g; F_(1,49)_ = 7.23; *p* < 0.01).

The administration of GSSG has improved (F_(1,49)_ = 4.05; *p* < 0.05) by 25% the plasma concentration of total glutathione (GSH + GSSG) (Figure 1). The efficient dose was 750 µg/kg/d. There was no difference between the lowest doses (up to 330 µg GSSG/kg/d; F_(1,49)_ < 0.35), or between the highest doses (750 and 1300 µg/kg/d) and control (F_(1,49)_ < 0.25), suggesting that higher doses of GSSG achieve a plateau in plasma glutathione.

**Figure 1 antioxidants-11-01956-f001:**
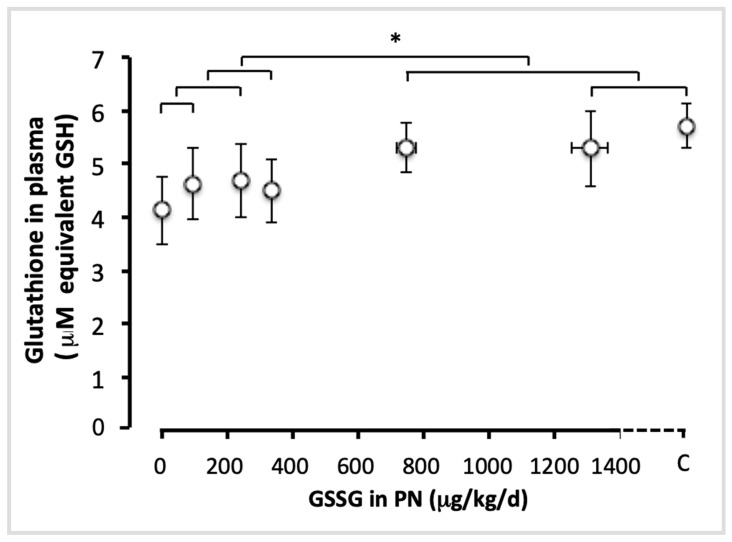
Total glutathione in plasma as a function of increasing doses of GSSG added to PN. The control (C) value was added to the abscissa after the highest dose of GSSG. Total glutathione: GSH + GSSG, expressed as GSH equivalent. The orthogonal comparisons were 0 vs. 95 µg/kg/d (no statistical difference), the two vs. 240 µg/kg/d (no difference), the first three doses vs. 330 µg/kg/d (no difference). The highest doses were not statistically different from the control: C vs. 1300 µg/kg/d was not statistically different, and both were not different from 750 µg/kg/d. However, the highest doses (750 and 1300 µg/kg/d) were statistically different (*p* < 0.05) from the lowest doses (0 to 330 µg/kg/d). Mean ± S.E.M. for plasma glutathione (*y* axis) and doses (*x* axis). The bars indicate the orthogonal comparisons; the absence of significant difference is shown by the absence of symbol above the bar; *: *p* < 0.05; n = 6–11 per group.

PN infusion with increasing GSSG concentrations significantly reduced the redox potential of glutathione in lungs (Figure 2). The efficient dose was 750 µg/kg/d. There was no significant difference between the first four tested doses (0–330 µg GSSG/kg/d; F_(1,49)_ < 3.4). However, the redox potential observed with the dose of 1300 µg/kg/d was close to the control value (F_(1,49)_ = 3.43; *p* < 0.07) but lower than the 750 µg/kg/d (F_(1,49)_ = 4.11; *p* < 0.05). Redox values in animals receiving the highest two doses of GSSG and in control animals were lower (more reduced) than those observed in animals receiving the lowest four doses (0–330 µg/kg/d; F_(1,49)_= 14.6; *p* < 0.001). Thus, a plateau in the redox values was not reached by the effective dose (750 µg/kg/d).

**Figure 2 antioxidants-11-01956-f002:**
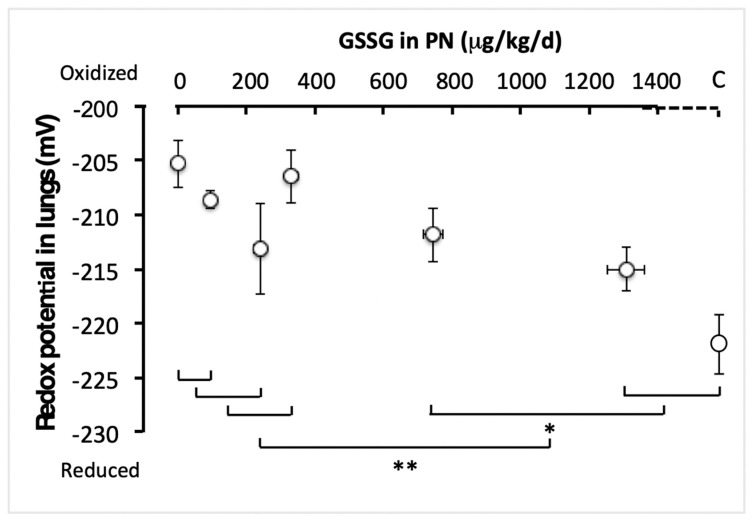
Pulmonary redox potential of glutathione as a function of increasing GSSG added to PN. The control (C) value was added to the abscissa after the highest dose of GSSG. The orthogonal comparisons were 0 vs. 95 µg/kg/d (no statistical difference), the two vs. 240 µg/kg/d (no difference), the first three doses vs. 330 µg/kg/d (no difference). The highest dose (1300 µg/kg/d) was not statistically different from the control, while the two were statistically different (*p* < 0.05) from 750 µg/kg/d. The highest doses (750 and 1300 µg/kg/d) and the control were statistically different (*p* < 0.001) from the lowest doses (0 to 330 µg/kg/d). Mean ± S.E.M. for redox values (*y* axis) and doses (*x* axis). The bars indicate the orthogonal comparisons; the absence of significant difference is shown by the absence of symbol above the bar; *: *p* < 0.05; **: *p* < 0.001; n = 6–11 per group.

Figure 3 shows the pulmonary levels of GSH for each group. There was no difference between doses ranging from 0 to 240 µg/kg/d (F_(1,49)_ < 1.6), nor those between 330 and 1300 µg/kg/d and the control (F_(1,49)_ < 1.4). However the difference between the lowest (0–240 µg/kg/d) and the highest doses (330–1300 µg/kg/d), including the control, was highly significant (F_(1,49)_ = 9.13; *p* < 0.001). The most efficient dose was 330 µg/kg/d. This dose reached a plateau in lung GSH content.

**Figure 3 antioxidants-11-01956-f003:**
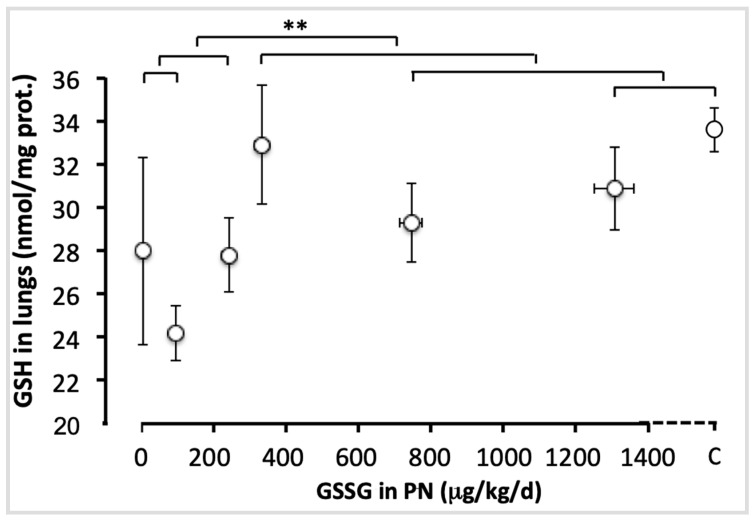
Pulmonary GSH levels as a function of increasing doses of GSSG added to PN. The control (C) value was added to the abscissa after the highest dose of GSSG. The orthogonal comparisons were 0 vs. 95 µg/kg/d (no statistical difference), both vs. 240 µg/kg/d (no difference). The highest dose (1300 µg/kg/d) was not statistically different from the control, the two were not different from 750 µg/kg/d, nor from 330 µg/kg/d. The highest doses (330 to 1300 µg/kg/d) and the control were statistically different (*p* < 0.01) from the lowest doses (0 to 240 µg/kg/d). Mean ± S.E.M. for GSH values (*y* axis) and doses (*x* axis). The bars indicate the orthogonal comparisons; the absence of significant difference is shown by the absence of symbol above the bar; **: *p* < 0.01; n = 6–11 per group.

The statistical model of comparisons shown in Figure 4 generated significant variations in GSSG values between animals with the four lowest doses. The GSSG level was the highest in animals with 330 µg/kg/d (F_(1,49)_ = 5.77; *p* < 0.05). There was no difference between the highest doses (750 and 1300 µg/kg/d) and the control (F_(1,49)_ < 1.6). GSSG levels were lower in animals with the two highest doses and control than in the four lowest doses (F_(1,49)_ = 18.2; *p* < 0.001). Thus, 750 µg/kg/d was considered as the efficient dose, allowing for a plateau to be reached.

**Figure 4 antioxidants-11-01956-f004:**
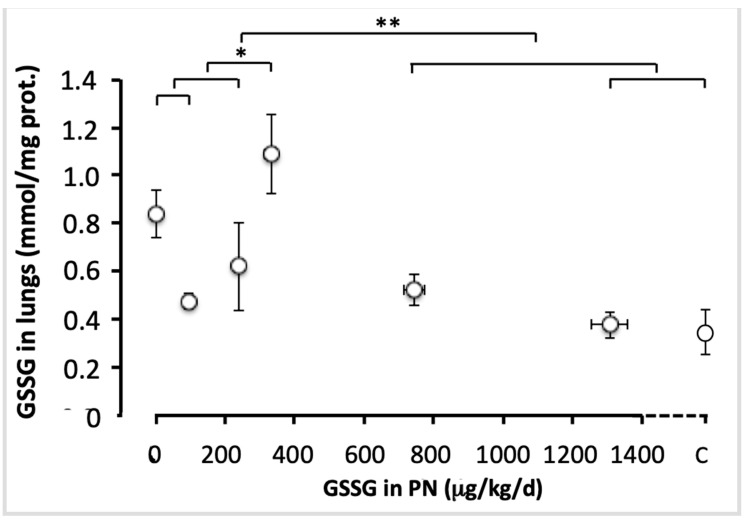
Pulmonary GSSG levels as a function of increasing doses of GSSG added to PN. The control (C) value was added to the abscissa after the highest dose of GSSG. The orthogonal comparisons were 0 vs. 95 µg/kg/d (no statistical difference: *p* < 0.07), both vs. 240 µg/kg/d (no difference). The first three doses were different from 330 µg/kg/day (F_(1.49)_ = 5.77; *p* < 0.05). Among the highest doses, 1300 µg/kg/d were not statistically different from the control, both were not different from 750 µg/kg/d. The highest doses (750 to 1300 µg/kg/d) and the control were statistically different (F_(1.40)_ = 13.7, *p* < 0.001) from the lowest doses (0 to 330 µg/kg/j). Mean ± S.E.M. for redox (*y* axis) and dose (*x* axis). The bars indicate the orthogonal comparisons; the absence of significant difference is shown by the absence of symbol above the bar; *: *p* < 0.05; **: *p* < 0.001; n = 6–11 per group.

Figure 5 shows representative pictures of a histological preparation for each group of animals receiving PN containing different amounts of GSSG, while Figure 6 shows the alveolarization index. The alveolarization index did not vary between the groups that received doses of 0 to 240 µg/kg/day (F_(1,49)_ < 0.1). The index did not vary either between the groups which received doses of 330 to 1300 µg/kg/day and the control group (F_(1,49)_ < 0.4). However, the difference between the lowest (0–240 µg/kg/d) and the highest doses (330–1300 µg/kg/d), including the control, was highly significant (F_(1,49)_ = 18.1; *p* < 0.0001). The most efficient dose was 330 µg/kg/d.

**Figure 5 antioxidants-11-01956-f005:**
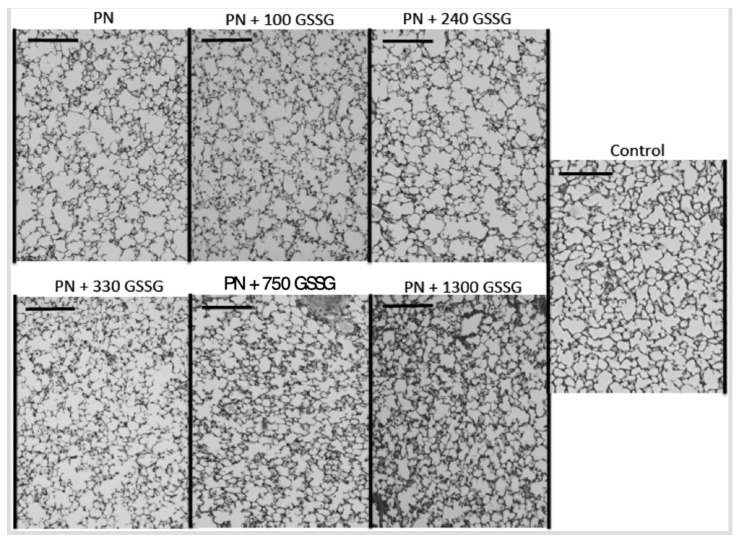
Representative pictures of a histological preparation of each group studied. Control: animals of the same age without any manipulation other than being fed with standard Guinea pig food; PN: parenteral nutrition; GSSG: dose of disulphide form of glutathione expressed in µg/kg/d. Bar on each sub-picture: 0.2 mm.

**Figure 6 antioxidants-11-01956-f006:**
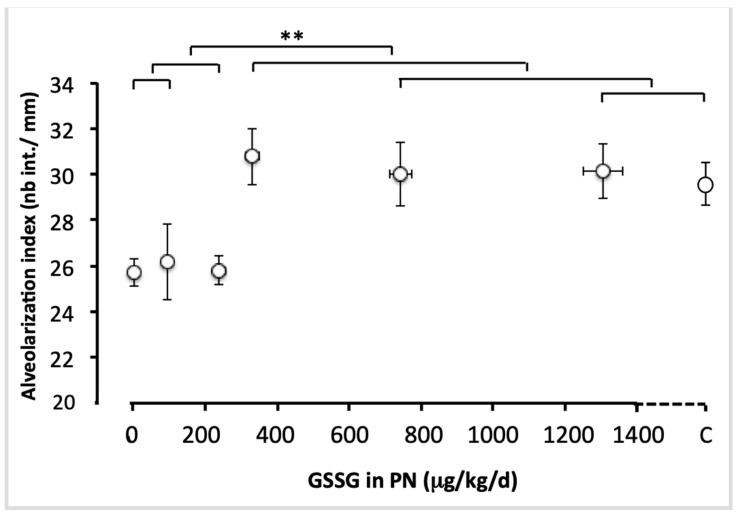
Alveolarization index as a function of increasing doses of GSSG added to PN. The control (C) value was added to the abscissa after the highest dose of GSSG. The orthogonal comparisons were 0 vs. 95 µg/kg/d (no statistical difference), both vs. 240 µg/kg/d (no difference). Among the highest doses, 1300 µg/kg/d were not statistically different from the control, the two were not different from 750 µg/kg/d nor from 330 µg/kg/d. The highest doses (330 to 1300 µg/kg/d) and the control were statistically different (F_(1.40)_ = 18.1, *p* < 0.0001) from the lowest doses (0 to 240 µg/kg/d). Mean ± S.E.M. for alveolarization indexes (*y* axis) and doses (*x* axis). The bars indicate the orthogonal comparisons; the absence of significant difference is shown by the absence of symbol above the bar; **: *p* < 0.0001; n = 6–11 per group.

GSSG levels are dependent on glutathione peroxidase (GPx) and reductase (GR). Thus, their activities were measured in selected groups, as shown in Table 1. There was no impact of GSSG addition in the PN, nor for GPx (F_(1,25)_ < 0.8) nor GR (F_(1,25)_ < 1.5). However, the GPx value in the control group was 15% lower (F_(1,25)_ = 7.4; *p* < 0.05) than that in the PN groups while the GR values did not differ between groups (F_(1,25)_ < 1.4).

## 4. Discussion

The main finding of the study is that supplementing PN with GSSG prevents hypoalveolarization, which has previously been associated with the presence of peroxides in PN [3,5,9,10,11]. The dose–response study proposes an effective GSSG threshold dose. The lower doses were ineffective, while higher doses yielded no additional benefit, with the alveolarization index reaching a plateau with a mean value similar to that seen in the control group. The results confirm the expected mechanism, in which infused GSSG allows for: (1) an increase in plasma glutathione concentration, (2) an increase in GSH levels in the lungs, (3) a reduction in the value of GSSG and (4) a shift in the redox potential value approaching that observed in the control group.

Parenteral nutrition is a significant source of oxidant molecules for premature newborns requiring this mode of nutrition [14]. The animal model has shown that these molecules, mainly peroxides, induce hypoalveolarization [3,9,10,11]. In humans, these peroxides are associated with BPD [5]. The oxidative stress induced by these molecules is a triggering event, leading to exaggerated apoptosis in the lungs [9,10]. GSH is the key molecule in the detoxification of these molecules. Unfortunately, glutathione levels are low in premature infants [5,20,21,22]. Therefore, the addition of glutathione in PN to improve the anti-peroxide capacity of newborns makes sense. Others have demonstrated the utility of glutathione supplementation to improve substrate levels in various animal models [23] for glutathione synthesis or to prevent the pulmonary toxicity of hyperoxia [24]. Our previous animal study confirmed the concept of protecting pulmonary integrity by adding GSSG in a PN solution [10]; the dose used corresponded to the highest dose used in the present study. In order to propose a clinical study to be carried out in extreme premature newborns, it was important to qualify the effective dose of GSSG to be used during this clinical study.

Conceptually, GSSG is used as a pro-cysteine molecule. The physiological source of cysteine for synthesis of GSH in tissues is glutathione in plasma [25]. The addition of glutathione has already been demonstrated to be effective in increasing the amount of cysteine [23,26]. The cellular availability of this amino acid is a limiting step for the synthesis of GSH [27,28]. In blood, γ-glutamyl transpeptidase transfers the γ-glutamyl moiety of glutathione to another amino acids in the circulation [29]. The dipeptides formed are taken up by cells, endothelial as well as lung epithelial cells, where membrane dipeptidases release cysteine, which thus becomes available for de novo synthesis of GSH [29,30]. GSH is the essential cofactor of GPx in the detoxification of peroxides. While we assume that its activity is adequate in premature infants [5,13], GSH levels are low due to cysteine deficiency. Indeed, the main way of obtaining this amino acid is the transformation of methionine into cysteine. The last step of this process is immature in premature infants [31,32], while peroxides from PN inhibit the first step [33]. One consequence is low glutathione in plasma. In our animal model, the plasma glutathione concentration was at 75% of the control value. In premature infants, the plasma glutathione is 25% of normal concentrations [5,22]. The parenteral administration of 48 µmol GSH/kg/h (corresponding to 350 mg/kg/d) to premature baboons improved their metabolism of cysteine and methionine, with an increase in the plasma concentration of cysteine to a similar concentration to that observed in adult baboons during their first week of life [26]. Here, the plasma glutathione deficiency was corrected with a dose that was approximatively 450 times lower, reaching a plateau with 750 µg GSSG/kg/d. The short plasma half-life of glutathione, as measured in humans, at 14 ± 3 min (mean ± S.E.M.) [34], limits the accumulation of glutathione, explaining the plateau observed here with the highest doses of GSSG, and possibly the plasma concentration of cysteine during the first weeks of life of preterm baboon infants [26]. However, Stabler et al. [26] reported a greater arterio-alveolar oxygen gradient at 1 week of life in premature baboons on GSH infusion compared to animals on cysteine infusion. The gradient was similar to that seen in premature baboons without GSH or cysteine treatment. There was no lower gas exchange in GSH-treated animals. An interaction between infused GSH and endogenous NO on vascular reactivity was hypothesized. In contrast, our study used GSSG, added in the parenteral nutrition solution, at a does that was three orders of magnitude lower. Nevertheless, in a phase I clinical study, particular attention must be paid to the oxygenation of premature infants.

Observing an increase in GSH and a decrease in GSSG in the lungs of animals receiving GSSG-enriched PN suggests that infused GSSG was used as a precursor of cysteine for the de novo synthesis of GSH, and that the peroxides were detoxified upon arrival in the lungs (lower generation of GSSG). This explains the shift in the redox potential to a more reduced state. This is consistent with reports of the association between oxidized redox potential and the severity of BPD [12] and its incidence [6] in premature infants. To detoxify peroxides, glutathione is essential as long as the GPx activity is sufficient. However, data on GPx activity in human preterm infants are few and inconsistent, relating only to the blood compartment. One can imagine that the ontogenesis of GPx activity is dependent on sex [35], gestational age (according to its needs during uterine development) and postnatal age, where oxidative stress, even just breathing air containing 21% oxygen, stimulates nuclear factors such as Nrf2, inducing its synthesis, provided the cell has a nucleus. Thus, this postnatal change may not take place in the erythrocytes in the first days of postnatal life, but could possibly occur in the lungs. A short review of the literature suggests that GPx erythrocyte activity is independent of postnatal age [36,37,38], while the impact of gestational age at birth is not certain. Indeed, some studies report an increase [3,36,39], while others show a decrease [40] or no change [37] in GPx activity.

Zima et al. [13] state that the activity of GPx in fetal blood was lower between 17 and 25 weeks of gestation than between 26 and 35 weeks, with similar values to that of the adult. Mohamed et al. [5] report plasma GPx activity in premature infants born at an average of 26 weeks, at similar levels to those reported in adults. While these results are conforting, the question remains as to whether the blood value accurately reflects the lung value. Animal studies indicate that, in Sprague-Dawley rat lung, activity increases in the first week of life [41], while in guinea pig lung, GPx mRNA levels increase during the last period of gestation before decreasing after birth [42]. The fact that the effective dose of GSSG to reduce the redox potential is twice that required to prevent lung remodeling suggests that it is not necessary to completely detoxify the peroxides, but to reduce their concentrations to below a specific toxic level. Even with a certain deficiency in GPx, a greater availability of GSH could promote sufficient activity to reduce the tissue level of peroxides below the threshold of toxicity.

The three-times-higher activity of GPx compared to that of glutathione reductase (Table 1) supports the concept that, in the presence of high peroxide levels, the generation of GSSG by GPx activity exceeds the ability of glutathione reductase to recycle to GSH. As the tissue level of GSH is normally close to the Km of the enzyme, with glutathione deficiency, GPx activity does not achieve its full measure. Thus, the detoxification rate of peroxides could be insufficient, allowing for them to accumulate [5]. At this peroxide level, GSSG generation increases. Therefore, by adding glutathione to PN, GSH synthesis in the lungs increases, supplying enough GSH to GPx to reduce peroxides as soon as they enter the lungs. The fact that, compared to the control, GPx activity was higher in animals receiving PN, suggests that the peroxides in the tissue are at a level that induces its synthesis. Even at the highest dose of GSSG, GPx activity remained higher than that of the control group, suggesting that oxidative stress has not been fully resolved.

A limitation of the study is the use of male animals. Since glutathione metabolism is central to the concept, and glutathione metabolism differs between males and females in preterm infants [43], it is important to stratify further clinical studies according to the sex of the infant. Another limitation of extrapolation to human preterm infants is the uncertainty of GPx activity in the lungs of this population. Even if GPx activity might differ between erythrocytes and lungs, we cannot ignore that Fu et al. demonstrated that erythrocyte GPx activity was approximately 30–35% lower at 5 days of life in preterm infants (from 25 to 30 weeks gestation) who developed BPD [44]. On the other hand, the best way to optimize this activity is to ensure that the concentration of substrate, for example, GSH, is not a limiting factor. Thus, even with lower GPx activity, premature infants would benefit from supplemental glutathione.

Historically, BPD was related to O_2_ exposure [45,46,47]. Recently, Mohamed et al. showed that O_2_ supplementation and peroxides contaminating PN are the two main oxidants to which preterm infants are exposed during the neonatal period [6]. Both oxidants induce the oxidation of redox potential and increase the incidence of BPD [6]. Knowing that, in vivo, free oxygen will first be converted in superoxide anion before becoming hydrogen peroxide [7], a relevant question could be whether the addition of GSSG to PN will also be beneficial to infants exposed to O_2_ supplementation. Brown et al. showed that glutathione supplementation attenuated oxidative lung injury and protected preterm rabbits exposed to 95% O_2_ [24].

## 5. Conclusions

In conclusion, a major finding of the study is the observation that the GSSG dose required to prevent lung integrity and correct pulmonary GSH (330 µg/kg/d) was lower than that needed to fill the plasma deficiency (750 µg/kg/d). Thus, in a clinical situation aiming to find an effective dose for administration to infants, the parameter to be monitored is the plasma level of glutathione. Our results strongly suggest that the GSSG dose that corrects the plasma deficit in glutathione will be effective in preventing the development of BPD. The glutathione metabolism is highly and tightly regulated, limiting the possibility of toxicity with a higher-than-necessary dose of GSSG. The similarities between premature infants and our animal model in glutathione metabolism (including that of methionine-cysteine), in addition to the oxidative stress, disease severity, and low toxicity (if any) of GSSG at the proposed doses, the enrichment of the PN of preterm infants with GSSG is attractive. The evidence suggests that the pulmonary glutathione metabolism, including GPx activity, would allow for preterm infants to benefit from GSSG supplementation in parenteral nutrition, although only a clinical study can confirm this.

## Figures and Tables

**Table 1 antioxidants-11-01956-t001:** Lung activities of Glutathione Peroxidase (GPx) and Glutathione Reductase (GR).

	GPx (nmol/Min/mg Prot)	GR (nmol/Min/mg Prot)
PN (no GSSG)	16.0 ± 0.7 ^a^	5.4 ± 0.4 ^c^
PN + GSSG (330 μg/kg/d)	15.8 ± 0.6 ^a^	5.4 ± 0.4 ^c^
PN + GSSG (1300 μg/kg/d)	15.1 ± 0.7 ^a^	4.8 ± 0.5 ^c^
Control	* 13.4 ± 0.8 ^b^	4.6 ± 0.4 ^d^

The presence of GSSG in PN had no effect on GPx (a vs. a: F_(1,25)_ < 0.8) or GR (c vs. c: F_(1,25)_ < 1.5). However, GPx value was lower in control than in PN groups (a vs. b: F_(1,25)_ = 7.4, *: *p* < 0.05). The GR values did not differ between PN groups and control (c vs. d: F_(1,25)_ = 1.4). Mean ± S.E.M. (n = 6 to 8 per group).

## Data Availability

All data is included in the article.

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
