# Peer review of "Dose–Response Effects of Glutathione Supplement in Parenteral Nutrition on Pulmonary Oxidative Stress and Alveolarization in Newborn Guinea Pig"

_antioxidants, 2022, doi:10.3390/antiox11101956_

Round 1
Reviewer 1 Report (Previous Reviewer 3)
Authors need to provide information on the statistical comparisons between the followings for each figure. Authors can then perform specific comparison strategies that they wish to perform.
C vs 0
C vs 95
C vs 240
C vs 330
C vs 750
C vs 1300
0 vs 95
0 vs 240
0 vs 330
0 vs 750
0 vs 1300
95 vs 240
95 vs 330
95 vs 750
95 vs 1300
240 vs 330
240 vs 750
240 vs 1300
330 vs 750
330 vs 1300
750 vs 1300
Author Response
Reviewer 1:
C 1: Authors need to provide information on the statistical comparisons between the followings for each figure. Authors can then perform specific comparison strategies that they wish to perform.
The reviewer suggests 21 comparisons (all possible comparisons).
A: When designing the study, a hypothesis and objectives were specified. To answer this, a set of experiments was designed and a statistical analysis plan was chosen. The use of an animal model forced us to take into account animal ethics which, among other things, requires using as few animals as possible. Our statistical analysis did not include multiple comparisons. Indeed, this method of comparison reduces the statistical power, leading to the use of a larger number of animals.
In this type of comparison, the same average of a specific group is used several times. Suppose that chance has erroneous this average. This error is repeated several times. To limit this bias, statisticians have developed statistical methods that take this possibility into account. For example, using the t test for the 21 comparisons as proposed by the reviewer, to conclude with an alpha error of 0.05, one must use a t value corresponding in the t table to an alpha value of 0.0024 (value alpha of 0.05 / 21 comparisons). Even using Dunnett's method which compares only one group to all the others, generating for us 6 comparisons, we are penalized with 0.05 / 6 = 0.0083.
For the present study, using multiple comparisons, the statistical power would be too low to demonstrate a difference. To increase statistical power, we need to increase the number of animals used. Having included the choice of multiple comparison as a statistical method when submitting the project to the ethics committee, I should have explained our choice knowing that this leads to a significant increase in the number of animals. Ethically, we are not able to offer statistical analyzes including multiple comparisons.
Our choice was to use an ANOVA which includes only orthogonal comparisons. This means that an average is only used once. So, for example, in the present study, the first comparison was between doses 0 and 95. The second comparison was between the 240 group and the new group which includes data from group 0 and 95, etc. These orthogonal comparisons reduce the possibility that a group's value was generated by chance. Combining this data with other data reduces the risk of error propagation. The ANOVA is built to take this risk into account on all the comparisons (here 6 because 7 groups). This mode of orthogonal comparisons gives freedom of possible comparisons. These comparisons must be consistent with the initial hypothesis. To test the hypothesis, each measured parameter must be analyzed to find the lowest dose that generates a different value (p < 0.05), this dose is defined as the effective dose. Subsequently, the effective doses of the different parameters would be discussed. After an effective dose is discovered by a preliminary analysis, comparisons will be chosen to validate whether this effective dose is part of a plateau, and the new average made up of the data from the effective dose, the higher doses and the control, would be compared to the average made up of the values obtained with the doses lower than the effective dose.
Therefore, since the initial experimental plan was to use an ANOVA whose comparisons are orthogonal, the sample size was fixed. With this number of animals, it would not have been possible to highlight differences between groups using multiple comparisons (statistical power too low).
In the Statistical Analysis section of the revised manuscript, the following text has been added: ‘After finding an effective dose by preliminary analysis, this dose was compared to the higher doses including the control (a plateau was suspected), and all doses below the effective dose were compared to the highest doses including the effective dose. This comparison made it possible to confirm or not that the effective dose has reached a plateau.’
Reviewer 2 Report (New Reviewer)
This new reviewer feels that all comments to previous reviewers were addressed. Additional experiments are not needed, but this reviewer does have two additional items that need to be addressed:
1) The GSSG production is profound in the presence of high levels of peroxides during the naturalization process, which leads imbalance in GSH to GSSG, and GSSG generation is also greater than the glutathione reductase to recycle such elevated GSSG to GSH. Please clarify how you would ensure that the infusion of GSSG did not aggravate the above reaction. Why does not the in vivo system convert such elevated GSSG into its constituent amino acids rather providing intravenous GSSG? If so, what is the role of γ-glutamyl transferase in those reactions, as both are similar or different?
2) Though the stability of GSSG is better than GSH, the level of GSSG also declines with time or temperature at NICU or incubation. It would be better to clarify the in vitro stability of GSSG in the PN just before administration.
Author Response
Manuscript Antioxidants-1897818 entitled: ”Dose-response effects of glutathione supplement in parenteral nutrition on pulmonary oxidative stress and alveolarization in newborn guinea pig”
Authors: Jean-Claude Lavoie *, Ibrahim Mohamed and Vitor Teixeira
Answers to the Reviewers.
Comment (C), Answer (A)
Reviewer 2:
This new reviewer feels that all comments to previous reviewers were addressed. Additional experiments are not needed, but this reviewer does have two additional items that need to be addressed:
C 1a: The GSSG production is profound in the presence of high levels of peroxides during the naturalization process, which leads imbalance in GSH to GSSG, and GSSG generation is also greater than the glutathione reductase to recycle such elevated GSSG to GSH. Please clarify how you would ensure that the infusion of GSSG did not aggravate the above reaction.
A: Of course, in vivo a high level of peroxides could lead to elevation of GSSG in cells. The ratio of GSSG to GSH must be tightly regulated as its is the mean factor affecting redox potential of the cell. This redox status has a great importance in cellular cycle (Schaffer and Buettner, 2001). The excess GSSG is recycled to GSH by glutathione reductase. However, the activity of GPx (favor generation of GSSG) was three time values than those of GR (see Table 1). Activities presented her have been measured at VMax. In vivo, the low glutathione induced by PN [9,10,14,16,17,29] limits the activity of GPx. In premature infants, the urinary concentration of ascorbylperoxide, which derived from PN, has been measure during the first week of life. The levels increased with time in infants who developed BPD [5]. The conclusion was that the infants who are not able to detoxify peroxides, ascorbylperoxide can reach a toxic level in lungs.
Hence, we can imagine that high level of peroxides from PN are not fully detoxified, peroxides accumulate. Even that activity of GPx is not optimal (caused by lower level of GSH), the high level of peroxides could favor a high generation of GSSG, as observed here at day 4 of the PN infusion. The hypothesis is that an improvement in pulmonary GSH content will produce better GPx activity, an activity that will prevent the accumulation of peroxides and the overproduction of GSSG.
GSSG does not cross the cellular membrane. Thus it cannot increase by itself the intracellular pool of GSSG. GSSG in circulation is processed by gGT anchored in epithelial cells. The final process is the release of amino acids, of which cysteine, inside the cell for a de novo synthesis of GSH. Our results demonstrate a reduction of GSSG in lungs of animals infused with the highest doses of GSSG.
The second paragraph of the Introduction already discusses this mechanism. A new paragraph has been added in Discussion section (second paragraph on page 11) of the revised manuscript.
C1b: Why does not the in vivo system convert such elevated GSSG into its constituent amino acids rather providing intravenous GSSG? If so, what is the role of γ-glutamyl transferase in those reactions, as both are similar or different?
A: The only enzyme to breakdown glutathione in its constituent amino acids is the Gamma-Glutamyl transpeptidase. This enzyme is present in lung (Ingbar DH, Hepler K, Dowin R, Jacobsen E, Dunitz JM, Nici L, Jamieson JD, . Gamma-Glutamyl transpeptidase is a polarized alveolar epithelial membrane protein. Am J Physiol 1995;269:L261-L271). However, the intracellular GSSG is not available to this enzyme as its is an ecto enzyme. It is anchored in cell membranes with active site bathed in the extracellular medium (Nash B, Tate SS. Biozynthesis of rat renal y-glutamyl transpeptidase. Evidence for a common precursor of the two subunits. J Biol Chem 1982; 257:285-288).
Because this interesting point is not the main subject of the study, no changes were made to the revised manuscript.
C2: Though the stability of GSSG is better than GSH, the level of GSSG also declines with time or temperature at NICU or incubation. It would be better to clarify the in vitro stability of GSSG in the PN just before administration.
A: A recent article [17] demonstrated that the loss of GSSG in PN is caused by the presence of cysteine in PN. The reaction leads to the generation of a mixed disulfide cysteine-glutathione. This molecule is recognized as a pro-cystine in vivo, which can be used to increase the synthesis of glutathione in the tissues.
This text has been added in the fourth paragraph, p.3, of the Experimental design.

Reviewer 3 Report (New Reviewer)
Lavoie et al. have performed an exciting and thought-provoking study. The group has been working on TPN and lung injury for a long time, contributing immensely to the literature. The study is interesting in that administering GSSG in PN would decrease BPD in premature neonates. The following are some of the suggestions to make the manuscript better.
Introduction –
What is the rate-limiting step in the glutathione system? Low GPx will lead to accumulation of peroxides in the body/tissues, as much as low GR not producing enough GSH. Is GPx central to the glutathione system in the lung? Is GR central to the accumulation of reduced glutathione?
There are studies on GPx in the lung (animal and cell culture studies). The authors have to cite these articles when they mainly talk about ‘studies on pulmonary GPx remain limited by clinical and ethical challenges. In addition, the authors have conducted an animal study. So, it is appropriate to cite literature on GPx in animals.
Methods –
The disulfide form of glutathione was added to PN. GSSG was added because of better stability in PN – If the hypothesis is that higher peroxides from PN produce more GSSG, why would you administer GSSG in PN? Would it not increase GSSG levels in the plasma, despite an increase in total glutathione?
Additionally, as you state in the introduction that ‘the rate of production of GSSG could be greater than the capacity of GR to recycle GSSG into GSH,’ the reviewer feels that administration of GSSG would increase the oxidized glutathione in the blood and tissues as well.
The authors cite that GSSG has an affinity for gamma-glutamyl transferase. Needs reference.
According to the authors, GSSG in PN is broken into constituent amino acids (where in the body?), and GSH is formed. This will not happen in blood. The reviewer wants the authors to be clear on how GSSG administered in PN would form GSH and increase reduced glutathione in the body (need reference). This is central to the hypothesis of the study.
The authors need to clear about the number of groups with the doses of GSSG in PN in the Methods section. It looks like the authors had the following groups – Control (No PN); PN+0/kg; (PN); PN+95/kg; PN+240/kg; PN+330/kg; PN+750/kg; PN+1300/kg). The dose of PN+1300 is PN+1350 in several places. Needs correction.
The reviewer feels that the comparison between groups is arbitrary. 0 vs. 95, 1350 vs. control, etc. We need repeated measures ANOVA or other statistics correct for multiple comparisons.
Results –
The reviewer is not aware of the following – F (1,49) = 7.23 or F(1,49) <1.6. The authors should explain something that is not used regularly in statistics. They need to define…where F(1,49) is so and so….
Figure 1. – Administration of GSSG improved plasma total glutathione. As they administer GSSG, the total (GSH +GSSG) is bound to increase. What is the purpose of this data? Additionally, the authors describe the efficient dose as 750/kg. The authors must explain how they defined the ‘efficient dose ‘and arrived at that dose. Again, what is the purpose of defining the ‘efficient dose’? This needs explanation in the methods section/discussion section.
Figure 2. The redox potential in the lungs – The control value has -220. So the lower the redox value, the better it is.
The authors are trying to establish that administering GSSG in PN at higher doses reduced redox potential in the lung. However, the reviewer is not sure how the authors demonstrated that relationship. It is also possible that higher GSSG administered elicited more inflammation, resulting in the body producing more GSH, resulting in lower redox potential.
Does GSSG elicit inflammation when administered by intravenous route?
Figure 3 – is buried in figure 4 in the manuscript. Hence cannot comment. However, from the figure legend – 1300 dose and control were no difference in GSH levels. Similarly, the GSSG in the lung was not different between the 1300 dose and control (figure 4). Therefore, the reviewer feels that the dose of GSSG in PN has no relationship with pulmonary GSH or GSSG.
Figure 5/6. An alveolar index is a simple tool as an indicator of lung structure. Is it possible to confirm one more piece of data that GSSG in PN improves alveolarization (an increase in lung elastin or decreases fibrosis – decreased lung collagen)? Also, the efficient dose varies from 750 (plasma) to 330 (lung). Why is that so?
Table 1. Lung GPx was not different between PN+gssg and PN groups. This is irrespective of no GSSG / 330 or 1300/kg dose. However, there was a slight decrease in the control group. How would the authors interpret the results? – Does GSSG administered has no role in the GSH system in the lung? High or no GSSG administered, did not change GSH / GSSG balance in the lung? In which case, how would they explain an improvement in the alveolarization index?
GPx is lower in the control group but not GR. Could it be that the administration of GSSG in PN resulted in a higher turnover of GPX by induction? As GR was not different among the groups and controls, is it possible that the load of GSSG in the lung was not high enough to induce GR among the groups?
Measuring hydroperoxides in the lung at different doses of GSSG administered with PN would be interesting in the present study.
Discussion.
The model the authors use is that of PN-induced lung injury. However, these animals are not ventilated or oxygen administered at any time. Are there studies on whether just TPN induces lung injury resulting in BPD in premature neonates? The authors should quote them to strengthen the analysis.
Author Response
Answers to Reviewers.
Comment (C), Answer (A)
Reviewer 3:
Comments and Suggestions for Authors
Lavoie et al. have performed an exciting and thought-provoking study. The group has been working on TPN and lung injury for a long time, contributing immensely to the literature. The study is interesting in that administering GSSG in PN would decrease BPD in premature neonates. The following are some of the suggestions to make the manuscript better.
C 1a: What is the rate-limiting step in the glutathione system? Low GPx will lead to accumulation of peroxides in the body/tissues, as much as low GR not producing enough GSH. Is GPx central to the glutathione system in the lung? Is GR central to the accumulation of reduced glutathione?
A: The action of these two enzymes is now discussed in the Discussion section. The following text has been added as second paragraph on page 11: ‘The three times higher activity of GPx than that of glutathione reductase (Table 1) supports the concept that in the presence of high levels of peroxides, the generation of GSSG by GPx activity exceeds the ability of glutathione reductase to recycle it to GSH. As the tissue level of GSH is normally close to the Km of the enzyme, in glutathione deficiency, the activity of GPx does not have its full measure. Thus, the rate of detoxification of peroxides could be insufficient, allowing them to accumulate [5]. At this level of peroxides, the generation of GSSG increases. Therefore, by adding glutathione to PN, GSH synthesis in the lungs increases, supplying enough GSH to GPx to reduce peroxides as soon as they enter the lungs. The fact that, compared to the control, the activity of GPx was higher in the animals receiving PN, suggests that the peroxides in the tissue are at a level which still induces its synthesis. Even at the highest dose of GSSG, the GPx activity remained higher than that of the control group, suggesting that oxidative stress has not been fully resolved.’
C1b: There are studies on GPx in the lung (animal and cell culture studies). The authors have to cite these articles when they mainly talk about ‘studies on pulmonary GPx remain limited by clinical and ethical challenges. In addition, the authors have conducted an animal study. So, it is appropriate to cite literature on GPx in animals.
A: Indeed, many other in vitro and animal studies exist. We agree, because it limits thinking about GPx to humans, the sentence "Pulmonary GPx studies remain limited by the clinical and ethical challenges of tissue collection" has been removed from the introduction. The importance of GPx, including the discussion of human and animal studies, is taken up, inter alia, in the 4th and 5th paragraphs, p. 14-15, from the Discussion section.
C2: Methods -
The disulfide form of glutathione was added to PN. GSSG was added because of better stability in PN – If the hypothesis is that higher peroxides from PN produce more GSSG, why would you administer GSSG in PN? Would it not increase GSSG levels in the plasma, despite an increase in total glutathione?
A: Indeed, GSSG in PN leads to an increase in GSSG in plasma. Epithelial cells are rich in gamma-glutamyl transpeptidase whose active sites are in the extracellular medium (plasma). This enzyme leads to the breakdown of circulating GSSG into its constituent amino acids, which are taken up and released into cells for de novo synthesis of the GSH. Our results on GSSG in the lungs support the concept. Increasing the dose of GSSG in the PN (and therefore in the plasma) induces a decrease in the level of GSSG in the lungs.
This concept is discussed in the fourth paragraph of the Experimental Design on page 3 and at the beginning of the second paragraph on page 10 of Discussion section.
No change was added to the revised manuscript.
C3: Methods -
Additionally, as you state in the introduction that ‘the rate of production of GSSG could be greater than the capacity of GR to recycle GSSG into GSH,’ the reviewer feels that administration of GSSG would increase the oxidized glutathione in the blood and tissues as well.
A: It makes sense that GSSG infusion increases the concentration of GSSG in the blood. But, physiologically, this GSSG cannot cross cell membranes. Also, too much glutathione in the blood is unlikely because its half-life is about 15 minutes (see second paragraph on page 10 in the Discussion section). This may explain that the dose effect of GSSG reaches a plateau in our study. The beneficial effect of infused GSSG is obligated by the action of gamma-glutamyl transpeptidase leading to the degradation of GSSG, as shown in the manuscript. No change was added to the revised manuscript.
C4: Methods -
The authors cite that GSSG has an affinity for gamma-glutamyl transferase. Needs reference.
A: Thank you to underline this mistake. Reference has been added in the revised manuscript.
C5: Methods -
According to the authors, GSSG in PN is broken into constituent amino acids (where in the body?), and GSH is formed. This will not happen in blood. The reviewer wants the authors to be clear on how GSSG administered in PN would form GSH and increase reduced glutathione in the body (need reference). This is central to the hypothesis of the study.
A: Thank you. The proof of concept underlining the present study has already been done and published. The purpose of the present study was to define the answer of each parameter measured here in function of the GSSG dose administered. The thinking was that the protection of lungs may require a different dose of GSSG than prevention of pulmonary oxidative stress or correction of plasma deficiency in glutathione. Our hypothesis was that we need to correct plasma deficiency to see a biological benefit. However, it was unclear whether this plasma glutathione-correcting dose was sufficient to preserve lung integrity. This data is important, because in the clinic it is only in the first days of life, during the administration of NP, that the clinician can assess the efficacy of the dose administered. Bronchopulmonary dysplasia is not diagnosed until several weeks later (at 36 weeks postmenstrual age). Thus, the comparison between the dose which corrects the plasma glutathione deficiency and the effective dose to maintain pulmonary integrity is useful. This provides the clinician with an easily accessible measure of efficacy of the dose used during treatment (during PN) of the newborn.
Thus, the biochemical mechanism linking infused GSSG and pulmonary GSH is described elsewhere. However, because the reviewer emphasizes the importance of re-describing these metabolic pathways, it is further described at the beginning of the first paragraph on page 10 of the Discussion section: ‘Conceptually, GSSG is used as a pro-cysteine molecule. The physiological source of cysteine for synthesis of GSH in tissues is glutathione in plasma [25]. The addition of glutathione has already been demonstrated effective in increasing the amount of cysteine [23,26]. The cellular availability of this amino acid is a limiting step for the synthesis of GSH [27,28]. In blood, γ-glutamyl transpeptidase transfers the γ-glutamyl moiety of glutathione to another amino acids in circulation [29]. The dipeptide formed is taken-up by cells, endothelial and lung epithelial cells, where membrane dipeptidases release free cysteine for a de novo synthesis of GSH [29,30].’
C6a: The authors need to clear about the number of groups with the doses of GSSG in PN in the Methods section. It looks like the authors had the following groups – Control (No PN); PN+0/kg; (PN); PN+95/kg; PN+240/kg; PN+330/kg; PN+750/kg; PN+1300/kg).
A: Seven groups were included in the study. Six of them received PN.
C6b: The dose of PN+1300 is PN+1350 in several places. Needs correction..
A: I don’t see the PN+1350 in the manuscript. If so, maybe it is because of a distortion during the transmission of the manuscript? Figures could be submitted separately if offered by the publisher.
C7: The reviewer feels that the comparison between groups is arbitrary. 0 vs. 95, 1350 vs. control, etc. We need repeated measures ANOVA or other statistics correct for multiple comparisons.
A: I refer this reviewer to my two-page response made for the first reviewer. This answer explains why the multiple comparison was not opportune for the study and how was determined the comparisons.
To use a repeated measures ANOVA, each animal would have had to be infused with the different doses of GSSG, which the experimental design did not allow.
Here is the answer for the first reviewer: When designing the study, a hypothesis and objectives were specified. To answer this, a set of experiments was designed and a statistical analysis plan was chosen. The use of an animal model forced us to take into account animal ethics which, among other things, requires using as few animals as possible. Our statistical analysis did not include multiple comparisons. Indeed, this method of comparison reduces the statistical power, leading to the use of a larger number of animals.
In this type of comparison, the same average of a specific group is used several times. Suppose that chance has erroneous this average. This error is repeated several times. To limit this bias, statisticians have developed statistical methods that take this possibility into account. For example, using the t test for the 21 comparisons as proposed by the reviewer, to conclude with an alpha error of 0.05, one must use a t value corresponding in the t table to an alpha value of 0.0024 (value alpha of 0.05 / 21 comparisons). Even using Dunnett's method which compares only one group to all the others, generating for us 6 comparisons, we are penalized with 0.05 / 6 = 0.0083.
For the present study, using multiple comparisons, the statistical power would be too low to demonstrate a difference. To increase statistical power, we need to increase the number of animals used. Having included the choice of multiple comparison as a statistical method when submitting the project to the ethics committee, I should have explained our choice knowing that this leads to a significant increase in the number of animals. Ethically, we are not able to offer statistical analyzes including multiple comparisons.
Our choice was to use an ANOVA which includes only orthogonal comparisons. This means that an average is only used once. So, for example, in the present study, the first comparison was between doses 0 and 95. The second comparison was between the 240 group and the new group which includes data from group 0 and 95, etc. These orthogonal comparisons reduce the possibility that a group's value was generated by chance. Combining this data with other data reduces the risk of error propagation. The ANOVA is built to take this risk into account on all the comparisons (here 6 because 7 groups). This mode of orthogonal comparisons gives freedom of possible comparisons. These comparisons must be consistent with the initial hypothesis. To test the hypothesis, each measured parameter must be analyzed to find the lowest dose that generates a different value (p < 0.05), this dose is defined as the effective dose. Subsequently, the effective doses of the different parameters would be discussed. After an effective dose is discovered by a preliminary analysis, comparisons will be chosen to validate whether this effective dose is part of a plateau, and the new average made up of the data from the effective dose, the higher doses and the control, would be compared to the average made up of the values obtained with the doses lower than the effective dose.
Therefore, since the initial experimental plan was to use an ANOVA whose comparisons are orthogonal, the sample size was fixed. With this number of animals, it would not have been possible to highlight differences between groups using multiple comparisons (statistical power too low).
In the Statistical Analysis section of the revised manuscript, the following text has been added: ‘After finding an effective dose by preliminary analysis, this dose was compared to the higher doses including the control (a plateau was suspected), and all doses below the effective dose were compared to the highest doses including the effective dose. This comparison made it possible to confirm or not that the effective dose has reached a plateau.’
C8: Results –
The reviewer is not aware of the following – F (1,49) = 7.23 or F(1,49) <1.6. The authors should explain something that is not used regularly in statistics. They need to define…where F(1,49) is so and so….
A: ANOVA generates the ratio of squared means between that of a source of variation (a comparison) to that of the residual. This ratio (denoted F) is compared to a critical F (0.05) which depends on the degree of freedom of the comparison and the residual. So for example F(1, 49) F is the value of the ratio and 1 is the degree of freedom of the comparison, and 49 is the degree of freedom of the residual.
Describing the F allows the reader to appreciate the level of significance of the comparison. However, if the editor recommends it, I can remove the F values from the text.
C9: Figure 1. – Administration of GSSG improved plasma total glutathione. As they administer GSSG, the total (GSH +GSSG) is bound to increase. What is the purpose of this data? Additionally, the authors describe the efficient dose as 750/kg. The authors must explain how they defined the ‘efficient dose ‘and arrived at that dose. Again, what is the purpose of defining the ‘efficient dose’? This needs explanation in the methods section/discussion section.
A: The definition of the ‘efficient dose’ is describe in the Statistical Analysis section as follows:
‘For each analyzed parameter, the objective was to use orthogonal comparisons making it possible to highlight the GSSG dose at the origin of a change in the initial values (in animals receiving PN without observed in animals receiving PN without GSSG. This dose was defined as the efficient dose.’
Compared to the dose required (330 ug/kg/d) to preserve lung integrity (Figure 6), the results in Figure 1 demonstrate that the animals need a higher level of GSSG (750 ug/kg /j) to correct plasma glutathione deficiency. Therefore, to test the efficacy of intravenous GSSG in premature infants, these results suggest that a dose that corrects plasma glutathione deficiency would be sufficient to prevent PN-induced lung damage. In a phase I clinical trial, the goal should be to find the dose that corrects plasma glutathione.
The following text has been added in Statistical Analysis section of the revised manuscript.
‘After finding an effective dose by preliminary analysis, this dose was compared to the higher doses including the control (a plateau was suspected), and all doses below the effective dose were compared to the highest doses. including the effective dose. This comparison made it possible to confirm or not that the effective dose has reached a plateau.’
In Results section, the following words has been added at the end of the second paragraph :
‘… suggesting that higher doses of GSSG achieve a plateau in plasma glutathione.’
C10a: Figure 2. The redox potential in the lungs – The control value has -220. So the lower the redox value, the better it is.
The authors are trying to establish that administering GSSG in PN at higher doses reduced redox potential in the lung. However, the reviewer is not sure how the authors demonstrated that relationship. It is also possible that higher GSSG administered elicited more inflammation, resulting in the body producing more GSH, resulting in lower redox potential.
A: The demonstration of the impact of GSSG doses on redox was make by the statistical analysis (see new text in Statistical Analysis section). At the end of the third paragraph in Results section, the following sentence has been added: ‘Thus, a plateau in the redox values was not reached by the effective dose (750 µg/kg/d).’
C10b: Does GSSG elicit inflammation when administered by intravenous route?
A: Thank you for the question. I don’t have the answer. However, we know that nuclear level of NFkB is increased in lungs of animals infused with peroxides that are present in PN (Elremaly W et al Redox Biol 2014). Reducing peroxide levels by infusing GSSG (in PN) must also reduce the activation of this important transcription factor of inflammation. Of course, in a clinical situation, inflammation markers must be monitored.
C11: Figure 3 – is buried in figure 4 in the manuscript. Hence cannot comment. However, from the figure legend – 1300 dose and control were no difference in GSH levels. Similarly, the GSSG in the lung was not different between the 1300 dose and control (figure 4). Therefore, the reviewer feels that the dose of GSSG in PN has no relationship with pulmonary GSH or GSSG.
A: I’m sorry for this difficulty with the figure. Infusion of GSSG had really improved the GSH content in lungs. Indeed, as reported in the fourth paragraph (beginning by ‘Figure 3 shows…’), for GSH, the efficient dose was 330 ug/kg/d and reached a plateau in lung GSH content.
C12: Figure 5/6. An alveolar index is a simple tool as an indicator of lung structure. Is it possible to confirm one more piece of data that GSSG in PN improves alveolarization (an increase in lung elastin or decreases fibrosis – decreased lung collagen)? Also, the efficient dose varies from 750 (plasma) to 330 (lung). Why is that so?
A: Of course, alveolar index is a simple tool of lung structure. The proof of concept of the beneficial impact on lung structure has been already published (Elremaly et al, FRBM, 2015; Elremaly et al, Redox biol. 2016). The aim of the present study was to discriminate the effective dose between the parameters. The infused peroxides must reach a toxic level before observing an increase in oxidative stress, an exaggerated apoptosis and a loss of alveoli. Indeed, photo-protection of NP which halves the peroxide level prevented these injuries in the animals. Additionally, Mohamed's study [5] on the link between BPD and urinary ascorbylperoxide concentration reported that premature infants who were unable to detoxify peroxides developed BPD. This study shows that the level of peroxide increases over time to reach a toxic level. Therefore, the hypothesis of the present study was that the infusion of GSSG works to reduce peroxides in the lungs to a non-toxic threshold. This dose of GSSG would be lower than that necessary to eliminate all the peroxides and thus protect the plasma pool of glutathione. Our results confirm this hypothesis. In our animal model, a dose of 330 ug/kg/d was effective in protecting the lung structure while a dose of 750 ug/kg/d was required to normalize plasma glutathione levels.
A large part of this answers has been added at the end of the Introduction in the revised manuscript.
C13: Table 1. Lung GPx was not different between PN+gssg and PN groups. This is irrespective of no GSSG / 330 or 1300/kg dose. However, there was a slight decrease in the control group. How would the authors interpret the results? – Does GSSG administered has no role in the GSH system in the lung? High or no GSSG administered, did not change GSH / GSSG balance in the lung? In which case, how would they explain an improvement in the alveolarization index?
A: I think that my answer at your first comment is appropriated here about the data presented in Table 1.
About questions on the roles played by infused GSSG on glutathione metabolism in lungs and on alveolarization, I think that the answer is included in all other comments by the present reviewer.
C14: Measuring hydroperoxides in the lung at different doses of GSSG administered with PN would be interesting in the present study.
A: I agree, but it is not possible because the samples were not stored under appropriate conditions.
C15: Discussion. - The model the authors use is that of PN-induced lung injury. However, these animals are not ventilated or oxygen administered at any time. Are there studies on whether just TPN induces lung injury resulting in BPD in premature neonates? The authors should quote them to strengthen the analysis..
A: A study by Mohamed [6] demonstrates that the prevalence of BPD is higher in premature infants receiving TPN for more than 2 weeks (supporting the concept of peroxide accumulation until a toxic threshold) even their FiO2 was lower than 25%. In fact, this article shows the additive effect of oxygen supplementation and duration of TPN.
That was already mentioned in the penultimate paragraph of the Discussion: ‘Recently, Mohamed et al. have shown that O2 supplementation and peroxides contaminating PN are the two main oxidants to which preterm infants are exposed during the neonatal period [6]. Both oxidants induce oxidation of redox potential and increase incidence of BPD [6].’

Round 2
Reviewer 1 Report (Previous Reviewer 3)
Please increase the number of animals.
Reviewer 3 Report (New Reviewer)
The reviewer thanks the authors for making appropriate changes to the manuscript.
One suggestion -
the p-value is the most commonly used statistic, and it is useful for the reader if the statistics are explained in p-value. F value may add more information; however, the addition of the p-value will simplify the study for the reader, which is the purpose of the research.
This manuscript is a resubmission of an earlier submission. The following is a list of the peer review reports and author responses from that submission.
Round 1
Reviewer 1 Report
I really enjoy the hypothesis and mission of this article, as improving redox state in preterm infants is essential, and few people are studying how to mitigate the negative impacts of TPN. So this topic is critically important for study. Also, I was convinced by the authors’ data that GSSG to TPN increased GSH in the plasma and lungs, as well improving the redox in the lungs.
- One of my comments is that several studies have found that glutathione peroxidase is lower in preterm infants than term infants/adults. The citation listed by the authors is a wonderful article about ascorbic acid and TPN, and reports GPx activity for preterm infants - but simply compares to adult reports and does not directly compare to term infants or adults. Several studies demonstrate that GPx activity is decreased in very preterm infants compared to term infants or adults DOI: 10.1111/j.1442-200X.2008.02662.x; DOI: 10.1002/(SICI)1097-0223(199612)16:12<1083::AID-PD994>3.0.CO;2-N; DOI: 10.1016/s0301-2115(02)00050-7. While in this article, GPX activity in plasma did not correlate with gestational age, this study was limited to babies 25-30 weeks and did not compare to term infants or adults. DOI: 10.1159/000112209. This is just plasma—there are also several preclinical studies demonstrating GPx increases in the lung across postnatal development. DOI: 10.1016/0167-4781(95)00214-6, DOI: 10.4067/s0034-98872007000700010; DOI: 10.1016/j.mce.2012.01.022. As GPx activity increased with hyperoxia in animal models, and one study reports is decreased in preterm infants who go on to develop BPD, I think the GPx story is incompletely understood and likely fairly nuanced. The manuscript could be enhanced by commenting on this, and adjusting lines 49-50 and again on page 9 which states GPx is similar in preterm infants compared to term infants and adults.
- Could the authors clarify if the pro-cysteine discussed in the introduction is the same TPN additive used in this study and being considered for a phase 1 trial?
- Could the authors comment on the weight loss for all the neonatal animals receiving TPN? As postnatal growth failure often correlates with alveolar development, were the authors surprised that they found improved alveolar development but no improvement in weight?
- For alveolarization I am less familiar with this method of reporting alveolar development. How do the authors pick the four lung fields of view- this seems low? I am more familiar with murine measurements where lungs are smaller but many more images are taken. Based on the n of 6-11 and the confidence intervals, I was also curious how much variability the authors found for this assessment and if the variability was impacted by individual animals GSH or redox values?
- Stylistically, the overall writing in the conclusion- some of their conclusions seem to indicate this dose may benefit infants or prevent BPD, including the last sentence.
- Last- could the authors comment on the study in preterm baboons where IV glutathione was given in TPN and an increase in OI and an increase in A-a gradient was observed? Stabler Am J Clin Nutr 2000; DOI: 10.1093/ajcn/72.6.1548
Reviewer 2 Report
Review: Dose-response effects of glutathione supplement in parenteral nutrition on pulmonary oxidative stress and alveolarization in newborn guinea pig
The manuscript submitted by Lavoie, Mohamed et Teixeira concerns an interesting topic. The role of oxidative stress in bronchopulmonary dysplasia (BPD) is indeed an essential part of the pathology that needs further and more focussed research.
The authors present data on the antioxidant effects of Glutathione disulphide on parental nutrition induced lung damage – due to peroxides in the nutrition – in newborn guinea pigs. Even though the data presented are certainly interesting, there are several reasons why this manuscript is not suitable for publication in its current form.
1) The authors use an animal model that cannot serve as a representative model for ‘bronchopulmonary dysplasia’. There is no prematurity – even though guinea pigs can be delivered up to 8 days prematurely and advanced alveolarization is already occurring in utero- which is almost present in all animal models (except rat and mouse, where they rely on hyperoxia). Even though they exclude this term in the title, BPD features explicitly in the introduction. As I read the methods section it seemed to me that newborn Guinee pigs (in the control group) were not even fed with formula milk. Is that even possible? Why were these controls not just left with the mother (as proper ‘external controls’)?
2) More than half of 32 references are self-citations or at least from the same research group. This is to say the least ‘indecent’. Many of the other references are very old citations (’80 or 90’s).
3) The statistical approach and the graphs are confusing (unclear which comparisons were done and it not mentioned whether a post-hoc test after the ANOVA was used). Moreover many data are presented in graphs, where the x-axis suggests a linear correlation, yet it should be used to represent different groups. In this case a larger table would be much more appropriate. In the graphs some of the statistical comparison (and significance) is depicted using a horizontal line. In some graphs some of these lines do not even correspond with actual data (e.g. some lines ‘finish’ above the number ‘1000 µg/kg/d’, which is not even a group in the study).
4) The evidence provided is not strong enough to allow for publication in this journal (e.g. histomorphometric analysis of lungs is insufficient, only basic evidence for pharmacological efficacy). Many more outcome measures are lacking: proper pulmonary morphometry (after pressure fixation of the lungs), lung function testing, inflammatory and developmental pathway analysis (using RNA or protein analysis), urinary peroxides…
5) Because of the mentioned reasons, the claims made in this manuscript are too strong (e.g. that the dose effective in term born guinea pigs provides will be the one that could be tested in extremely preterm human neonates). I sincerely hope that this study is not the basis for dose calculation in a human study as has been suggested in the manuscript.
Reviewer 3 Report
Abstract: “Hypothesis: this dose is the one that corrects the deficiency in plasma glutathione.” It is not clear what this means.
Abstract: “Control: animals without manipulation, orally fed.” Please make this a complete sentence.
Abstract states: “The effective threshold dose to improve pulmonary GSH and prevent the loss of alveoli was 330 μg/kg/d.” Fig. 3 does not look like there is statistical significance between 0 and 330 doses with large error bars. In Fig. 6, I don’t see a line between 0 and 330 doses that indicates statistical significance.
In all the figures, it is not clear what the bars refer to. For example, in Fig. 6, the bar at the top with ** seems to start at a point somewhere between 100 and 240 and end somewhere between 750 and 1300.
In summary, this reviewer cannot evaluate this manuscript without clarification of the graphs in terms of statistical designations.